# Liquid Biomolecular Condensates and Viral Lifecycles: Review and Perspectives

**DOI:** 10.3390/v13030366

**Published:** 2021-02-25

**Authors:** Temitope Akhigbe Etibor, Yohei Yamauchi, Maria João Amorim

**Affiliations:** 1Cell Biology of Viral Infection Lab, Instituto Gulbenkian de Ciência, 2780-156 Oeiras, Portugal; etibor@igc.gulbenkian.pt; 2School of Cellular and Molecular Medicine, University of Bristol, Bristol BS8 1TL, UK; yohei.yamauchi@bristol.ac.uk

**Keywords:** LLPS, viral factories, liquid organelles, viruses, biomolecular condensates, HIV, SARS-CoV-2, measles, vesicular stomatitis virus, influenza A virus, rabies

## Abstract

Viruses are highly dependent on the host they infect. Their dependence triggers processes of virus–host co-adaptation, enabling viruses to explore host resources whilst escaping immunity. Scientists have tackled viral–host interplay at differing levels of complexity—in individual hosts, organs, tissues and cells—and seminal studies advanced our understanding about viral lifecycles, intra- or inter-species transmission, and means to control infections. Recently, it emerged as important to address the physical properties of the materials in biological systems; membrane-bound organelles are only one of many ways to separate molecules from the cellular milieu. By achieving a type of compartmentalization lacking membranes known as biomolecular condensates, biological systems developed alternative mechanisms of controlling reactions. The identification that many biological condensates display liquid properties led to the proposal that liquid–liquid phase separation (LLPS) drives their formation. The concept of LLPS is a paradigm shift in cellular structure and organization. There is an unprecedented momentum to revisit long-standing questions in virology and to explore novel antiviral strategies. In the first part of this review, we focus on the state-of-the-art about biomolecular condensates. In the second part, we capture what is known about RNA virus-phase biology and discuss future perspectives of this emerging field in virology.

## 1. Introduction

Viruses are obligatory intracellular parasites highly dependent on the machinery of the hosts they infect. Inside cells, the adaptive plasticity of viruses to their cellular milieu is notable [1,2,3]. Seminal studies showed that viral adjustments to the cellular environment comprise sophisticated strategies for co-opting, readapting, and antagonizing different cellular pathways to facilitate viral replication, escape immune-related surveillance, and perpetuate viral transmission [4,5,6]. Many viral lifecycles were shown to remodel cellular architecture to form specialized platforms for viral genome replication and/or for the assembly of progeny neo-virions. Such platforms have gained different names over the years, depending on their function, adopting the names of “viral factories, viral inclusions, virosome, viroplasm, mini-nuclei and aggresome” [1,3,7,8]. Many viral platforms are intimately connected to membrane-bound surfaces such as endoplasmic reticulum (ER) (for hepatitis C virus, dengue, severe acute respiratory syndrome coronavirus-2 (SARS-CoV-2)), mitochondria (flock house virus), lysosome (Semliki Forest and rubella virus), peroxisome (tomato bushy stunt virus), and Golgi complex (Kunjin virus) [1,3,9,10,11,12,13,14]. Notable reviews focused on interactions between viruses and cellular structures exploring how viruses interact with membrane-bound organelles, remodel their organization, and alter their function to facilitate the production of progeny virions [1,3,9,10,11,12,13,14]. These studies provided canonical understanding of the biochemistry and physiology of the viral interplay with membrane-bound organelles and have taught us that viruses co-evolved processes to impact on every aspect of host cellular organization and structure investigated thus far. 

Recently, another layer of complexity—the physical properties of molecules in the cell—emerged as a key determinant in cellular organization. It was demonstrated that cells can achieve active and functional compartmentalization without recurring to membrane-bound organelles by forming what are now collectively known as biomolecular condensates [15,16,17,18]. Some of these compartments are obtained by a liquid–liquid phase separation (LLPS) mechanism, similar to the process as we understand for a water and oil mix. As weak and transient interactions become prevalent between selected multivalent and/or intrinsically disordered proteins (IDP) and their network, LLPS originates and becomes impenetrable to the remaining molecules of the milieu. Compartments that form this way are active, biochemically functional, and regulated in a multitude of ways, displaying the properties of liquids [15]. They have been observed in many cellular contexts including bacteria, higher eukaryotes, yeast, and archaea [15,19,20,21]. This shift in paradigm adds layers of complexity to the cellular structure, and it offers new possibilities to the understanding of how organisms work, control reactions, and adapt to stimuli. 

As contemporary biology refocused to shed light on the biophysical properties and principles for organizing membraneless organelles, it is now important to explore how viruses communicate with biomolecular condensates. Evidence shows that viruses co-opt membraneless organelles to facilitate viral replication and genome assembly, using parallel strategies to cellular systems, and this is the focus of the present review. Due to the growing number of manuscripts in this new field in virology, we now know that some replication compartments are liquid organelles involved in concentrating and assembling viral material. Examples include Negri bodies (NB) of rabies virus (RABV) infection, viral inclusions of measles (MeV), vesicular stomatitis virus (VSV), SARS-CoV-2 replication factories, etc. [22,23,24,25,26,27,28,29,30,31]. An extensive body of evidence shows that viruses can perform steps of the viral lifecycle in biomolecular condensates built *de novo* [24,27,29,32], but it can also interfere with existing ones. In fact, viruses were shown to interfere with the function and morphology of stress granules and processing bodies, which function as hubs for host translational shutoff and interfere with innate immunity signaling. These include nuclear factor kappa-light-chain-enhancer of activated B cells (NF-kB) or myxovirus resistance protein A (MxA) signaling pathways [33,34,35,36]. Therefore, there is a lot to learn on virus-phase biology, and it is paramount to shed light on the novel biophysical principles used by viruses to better understand viral lifecycles and discover opportunities to interfere with viral production. This review will focus on the state-of-the-art *de novo* biogenesis, maintenance, maturation, role, and principles of membraneless biomolecular condensates by RNA viruses. We will also discuss open questions and provide perspectives on the mechanisms of virus-based LLPS biology.

### 1.1. Phase Separation of Membraneless Organelles

The first part of the review explains the basic principles of LLPS and its consequences for biological systems. Our aim is to introduce new concepts to show that we need to re-wire our way of thinking to fully explore how LLPS principles can provide a new layer of understanding on host–pathogen interactions. It will become clear that the concentration of a molecule (and its composition) are critical parameters to consider. As a consequence, viral strategies to modulate cellular pathways to concentrate molecules at precise cellular locations gain more relevance than anticipated. In addition, we expose how the material properties of these compartments, liquids-, gel-, or solid-like, are tailored to suit specific functions. The second part of the review provides some examples of biomolecular condensates observed in cells infected by different viruses and explores what we know so far about them, referring to the caveats and discussing future ways to address them. 

The last decade earmarked a new era of LLPS in cell biology. Phase separation has been classically exemplified in biology using a well-mixed solution of oil and water, as an example [37]. In this solution, the system seems to be in one phase if well mixed; however, when left unperturbed, the oil and vinegar begin to separate/demix into two phases containing oil in one phase and water in another [37]. Since phase separation thermodynamically favors homotypic interactions over heterotypic ones, the stronger interactions of intra-water or intra-oil molecules over inter water–oil molecular interactions promotes the demixing of oil and water into two phases of oil and water [37]. The application of this concept to biology has several implications to our understanding of cellular processes.

Accumulating evidence suggests that condensates self-assembled by LLPS originate compartments permeable to selected components that quickly exchange material with its surroundings [15,16,37,38,39,40,41,42]. These biomolecular condensates have properties of liquids i.e., they divide and fuse, rearrange quickly internally, are highly dynamic and respond to stress and stimuli [16,37,43]. At the interface of the resulting phases, a non-membrane delimited phase boundary is formed for the selective permeability of homotypic (similar) molecules (and their interactome) while excluding heterotypic (dissimilar) molecules [38,40,44].

With the increased understanding on the formation of LLPS compartments, cytoplasmic bodies such as P-granules (germ body), stress granules, as well as nuclear structures like the Balbiany body, nucleolus, nuclear speckles, and nuclear paraspeckles became considered active cellular organelles with the properties of selective permeability. The formation of these organelles is regulated in ways that are still being understood, but it is clear that their deregulation can result in pathological conditions. This has elegantly been shown for proteins associated with neurodegenerative disorders by revealing mutations originating liquid-to-solid transitions and the formation of aggregates [17,45]. 

### 1.2. Concept 1: The Concentration of Molecules Drives the Formation and Demixing of Liquid Organelles

In the context of biological systems, biomolecular condensates are formed by hundreds of distinct proteins and RNA components [46,47]. However, for simplicity, in this review we explain the principles of LLPS for a binary system (see Figure 1 and its legend) to illustrate how concentration is a key variable driving the formation and demixing of phase-separated compartments.

It must be clearly understood that the rules of thermodynamics do not exactly translate into biology, and important examples are starting to emerge [50]. Biological systems are very far from being in thermodynamic equilibria, and this lack of equilibria is an absolute requirement for life. Not only is the composition of the cell and liquid compartments complex but it is also extremely variable, oscillating and adapting in response to several stimuli (e.g., nutritional state, several types of stress including microbes, extracellular signaling) [51,52,53,54]. Nevertheless, some general principles are being understood regarding the intrinsic properties of the components able to self-assemble into biomolecular condensates and the type of interactions they establish and even how biomolecular condensates may be kept away from equilibria or mature with time. These have important consequences for cellular structure and organization and will be explored in the following sections.

### 1.3. Concept 2: The Concentration of Molecule A in the Two Phases Can Be Orders of Magnitude Apart without the Assistance of a Physical Barrier

The critical concept of LLPS biomolecular condensates that constitutes a new framework in biology is that the concentration of molecule A in the two phases can be orders of magnitude apart without the assistance of a physical barrier (Figure 2). Importantly, reaching this difference in concentration does not require energy, as compartmentalization through LLPS is a spontaneous process, and so is its maintenance or transport across the boundaries [40]. In such a two-phase system, diffusion does not have to occur from higher to lower concentration, because the difference in the chemical potential at the interface is zero [16]. The materials that define the two phases might have different properties including distinct dielectric constants and affinities for the solute molecules. As a result, the interior of a condensate can be constituted by a solvent closer to an organic solvent rather than displaying aqueous properties and partition certain components while excluding others [55]. The consequences of this differential behavior are that these compartments can achieve an exquisite level of selection for partitioning molecules inside. For example, in vitro assembled DEAD box helicase 4 (Ddx4) compartments absorb single or hairpin-containing RNAs but exclude double-stranded RNA [55], which means that a fine-tuning in selection for inclusion in LLPS compartments is achieved. Furthermore, the formation of biomolecular condensates is reversible, and liquid organelles can assemble and dissolve in response to very small changes in conditions, including concentrations of components, ions, post-translational modifications (PTMs), and nucleic acids that result from environmental conditions or integrated cell signaling [56,57,58,59,60,61]. This leaves the cell with the potential to form functional biomolecular condensates under certain conditions and dissolve them when they change, responding quickly to metabolic alterations or to stress. Examples include how related enzymes can cluster to form the purinosome [61,62] and how their enzymatic products dissolve these structures or how stress granules form/disassemble into liquid organelles in response to signaling [51,63,64,65]. This provides an efficient means for regulating reactions in response to variations in conditions or for rearranging structures using a minimal investment of energy or allocated resources.

### 1.4. Concept 3: The Intrinsic Properties of Components and Interactions amongst Them Dictate the Formation and Properties of Phase-Separated Compartments

Biomolecular condensates were described in the early 1900s by many biologists who viewed the cytoplasm as a mixture of liquids in a form of emulsion [66]. Over decades, phase transitions were evoked several times both to describe “lipid rafts” [67] as well as cellular micro-compartmentalization [68]. Recently, Brangwynne et al. proposed that RNA and protein-rich P granules are asymmetrically distributed to the posterior end of the one-cell embryo by processes that resembled the condensation and dissolution of liquids [15], and this mechanism is essential to establish the first germ cell of Caenorhabditis elegans. The initial observation of the liquid behavior, including a spherical shape and the ability to fuse and divide, exchange material within, and react fast to stimuli [18], set in motion a series of seminal studies to explain the driving forces of condensate formation. It became established that the formation of biomolecular condensates depends on the properties of their constituents that are normally proteins and nucleic acids, mostly RNA, in still poorly defined compositions, that interact weakly and transiently with each other whilst avoid interacting with the milieu [16,18,69]. The molecules that reside in biomolecular condensates are of two types: molecules that drive their formation (drivers or scaffolds) and molecules that through interaction with the drivers/scaffolds are dragged to the condensates (clients or residents) [18,57]. In a series of seminal reports, the rules governing the driving forces and properties of condensate formation are beginning to be understood using LLPS models adapted from the field of associative polymers and hence considering macromolecules as polymeric chains. Examples include fused-in-sarcoma (FUS), TATA-binding protein-associated factor 2N (TAF15), Ewing sarcoma breakpoint region 1 (ESWR1), and heterogeneous nuclear ribonucleoprotein A1 (hnRNPA1) analyzed in light of the stickers-and-spacers framework [38,40,49,70,71,72,73]. 

Drivers of LLPS compartments comprise multivalent proteins and/or intrinsically disordered proteins/regions/domains (IDPs/IDRs/IDDs) that have low complexity and can also be called low-complexity domains (LCD) or prion-like domains (PrLD) [44,57,73,74,75,76,77,78,79,80]. Multivalent proteins are composed of N repeats of sticky domains/motifs that provide N-specific interactions [69]. Sticky interacting domains or stickers, that correlate with valency (number of ligations established), can be short linear motifs such as those found in IDP or folded domains such as for example RNA recognition motifs, SRC Homology 3 (SH3), PDZ (an initialism combining the first letters of the first three proteins discovered to share the domain—post synaptic density protein (PSD95), Drosophila disc large tumor suppressor (Dlg1)), and zonula occludens-1 protein (zo-1) or SUMO (small ubiquitin-like modifier) [81]. Stickers control the ability to form biomolecular condensates. They are composed of charged and aromatic amino acids, especially arginine and tyrosine. Stickers preferentially interact with other stickers in homotypic and heterotypic interactions [38,49,70], rather than interacting with the solvent, and thus establish high-order networks that are key in driving condensation [40,59,75]. Stickers determine the Csat mentioned in the phase diagram of Figure 1A in an inverse proportional manner: more stickers, lower Csat.

Other important contributors of phase separation are regions in between the sticky domains or spacers. These regions provide flexibility and structure to the network and can be structural domains or IDRs of low complexity. Space residues comprise glycine, serine, and glutamine [72,73]. Spacers are responsible for the material properties of the biomolecular condensates, as changing these residues leads to changes in diffusion and across phase boundaries of condensates [72,73]. Changes in glycine or serine to alanine result in the hardening of condensates and the change in glutamine to glycine results in increased fluidity [73]. Spacers control the rate at which cross-links are established and broken, which is a function of the number of possible ligations on space residues [38,70].

The types of interactions shown so far to permit phase separation include electrostatic, dipole–dipole, cation–π, and π–π interactions. Having cationic and aromatic stickers allows attracting molecules at a long range but also establishing specific cohesive interactions at a short range. As such, stickers can be said to encode the interaction range and strength in biomolecular condensates [82]. These interactions are possible on specific types of amino acids and nucleic acids prevalent on the types of proteins mentioned above [39,40,83,84]. Due to their length, flexibility, and multivalency, RNAs have been demonstrated to be a critical scaffold of proteins bearing RNA recognition motifs (RRM) for controlling phase separation [85]. However, their behavior is different depending on the identity of the biomolecular condensates (in some cases helping the formation, by enabling nucleation or establishing networks and in other cases permitting dissolution by electrostatic charge inversion, for example) [41,60,65,73,85,86,87,88,89,90,91]. Several manuscripts describe the role of RNA in promoting phase separation at a certain concentration, but if passing a critical concentration result in reentrant phase transitions. This means that after a concentration threshold RNA promotes the dissolution of biomolecular condensates, which offers an opportunity for a feedback loop in which the reaction products-in this case RNA-controls the formation or disassembly of biomolecular condensates, as observed in ribosomes or spliceosomes [58,87,92,93,94]. Phase separation also depends on the properties of the solvent in the intracellular milieu, which needs to disfavor solute–solvent interactions in order to drive spontaneous condensation.

### 1.5. Concept 4: The Formation of Phase-Separated Compartments Is Regulated

There are many other molecular modulators that are able to alter the phase threshold of molecules in in vitro reductionist models. These include chaperones, ATP (energy) [52], pH [95], ionic strength (salt concentration) [96], temperature [82], and PTMs (phosphorylation, ubiquitination, SUMOylation, methylation, acetylation, etc.) [54,82,97,98,99,100,101,102]. In particular, PTMs may either decrease or increase the Csat. Examples are: the phosphorylation of nephrin decreasing the condensation of Neural Wiskott-Aldrich syndrome protein (N-WASP) and non-catalytic region of tyrosine kinase (NCK) [103]; the arginine methylation of Ddx4 increasing the Csat of Ddx4 [82]. A particularly interesting example is the capacity of membranes and endomembrane surfaces to lower the phase concentration threshold. These may either restrict movement (from 3D to 2D) or increase the valency of components in order to generate LLPS nucleation sites, including actin assembly at membrane surfaces [97,99], synaptic vesicles in synapsin clusters [104], membrane contact sites [105], and a linker for activation of T cells (LAT)-based network of Ras signaling [106]. This offers an exquisite communication that lacks understanding in how the endomembrane system in the cells and biomolecular condensates regulate each other and for which readers are remitted to Zhao et al. for a review [107].

### 1.6. Concept 5: Biomolecular Condensates Adopt a Series of Material Properties and Undergo Phase Transitions or Mature Over Time

Biomolecular condensates formed by LLPS can opt a range of fluidities from liquid to gels and hydrogels where the constituents have reduced mobility, or to even solid inert fibers [38] (Figure 3). Interestingly, the function of biomolecular condensates has been postulated to be intimately associated with the material properties and their mobility: a condensate needs to have highly dynamic components for operating as signaling hubs or reactions centers, whilst elasticity and a degree of rigidity is important when condensates function as scaffolds. However, damaging phase transitions can also occur, leading to maturation or malfunction [108,109,110]. These result from changes in concentration, conformation, and available amino acids for engaging in interactions, or in cellular conditions able to alter the strength or type of interactions or of surface tension. This can occur via mutations, PTM, or aging, as observed in neurodegeneration [109]. For example, in the absence of shear force liquid droplets can fuse, leading to a coarsening of condensed phases to reduce energy, which is a process known as Ostwald ripening [111]. This process can lead to the maturation of liquid droplets into arrested gel, glass, or solid-like structures, presumably because of the establishment of more interactions, which in turn, results in the loss of flexibility of components [109]. Depending on the conditions, a liquid organelle can mature into a less dynamic gel/glass or into more stable solid-like structures by liquid–gel or liquid–solid transitions, respectively. In material physics and soft matter biology, gel/glass can also mature into solid crystals. This is exemplified in the polymerization of actin filament from a liquid Nephrin-Nck-N-WASP condensate [99] and aberrant neuropathological aggregates of FUS in amyotrophic lateral sclerosis (ALS) and frontotemporal dementia [45]. In the context of viral infection, MeV inclusion bodies have been shown to mature from liquid to a gel-like structure as infection progresses [26], and the impact that such changes have on function is unknown.

### 1.7. Concept 6: Biomolecular Condensates Are Functional and Regulated by Changing Valency or Strength of the Interactions Established amongst Drivers and Clients

The cell is a chaotic crowd of biomolecules including proteins, nucleic acids, carbohydrates, and lipids, which are organized into either membrane-bound organelles or collectives of separated biomolecular condensates that differ in composition, material state, and localization. The presence of a phase boundary allows biomolecular condensates to behave as functional compartments [18], to select components and buffer molecular noise [112]. The cellular functions of these structures have been described in a growing number of papers. For example, phase separation has been shown to promote the efficient spatiotemporal organization of cellular biochemistry, tune and accelerate biochemical reactions, modulate signal transduction, maintain proteostasis, nuclear structure, and the regulation of nucleic acid, sequester molecules for storage, and respond to cellular stress, thereby facilitating cellular fitness [55,56,65,80,99,103,112,113,114,115]. Given that viral infections take advantage of every aspect of cellular structure and organization, and that controlling metabolism, antiviral response, and biochemical resources are essential aspects for successful viral replication, it is expected that viruses take advantage of this type of compartmentalization. Advances made so far in understanding the link between soft matter science and virology will be the focus of the rest of the review.

## 2. Liquid Compartments in Viral Infections

Overview: Recently, many viruses were shown to assemble biomolecular condensates with the properties of liquids [5,22,23,24,25,26,33] (Figure 4), but our knowledge of liquid biomolecular condensates and especially of phase separation in viral infections is still in its infancy. In light of recent findings, such discovery suggests a change in paradigm on how viruses organize biochemical reactions. Despite being early days, compartments with liquid properties have been suggested to play roles in every step of viral lifecycles: viral entry, genome replication, assembly, and viral packaging (Section 2.1, Section 2.2, Section 2.3 and Section 2.4). Since these are essential for production of viral progeny, the perturbation of these steps could potentially abrogate viral infections, and hence, provide novel antiviral strategies. For each viral family/virus, we provide a small introduction explaining the proteins codified by each virus, and its lifecycle, because the characteristics of the proteins and the interactions they establish govern the ability of proteins to undergo LLPS, the material properties of liquid organelles, and the role they play in infection.

### 2.1. Virus Entry and Uncoating—The Case of Influenza A Virus

To establish infection, a virus must first attach, enter the host cell, and deliver the viral genome to the site of viral replication. This can be in the cytosol, the nucleus, or on cytoplasmic membranes [116]. Cellular and viral cues trigger virus entry and uncoating—they come from enzymes, chemicals, and receptor proteins that act directly to alter viral structure and infectivity. Enzymatic cues are often quality control machineries such as the ubiquitin proteasome system (UPS), endoplasmic reticulum-associated protein degradation (ERAD), and components that regulate LLPS. Chemical cues are provided via low pH or ions present in the endosomes or cytosol such as K+ and Ca2+. Receptor-mediated cues act directly on the virus and come from lipids, proteins, and sugars that are associated with membranes (plasma membrane, endosomes, nuclear pore complexes) but can also derive from the cytoskeleton e.g., microtubules, actin filaments, and their associated motor proteins dynein and kinesin [116].

Influenza A viruses (IAV) infect mammals, birds, and cause seasonal respiratory illness in humans and occasional pandemic outbreaks. IAVs are members of the Orthomyxoviridae family that contain four genera of influenza viruses, which are named A–D. The *Orthomyxoviridae* also includes *Thogotovirus*, *Isavirus*, and *Quaranjavirus* genera [117]. IAVs are enveloped viruses with the genome composed of single-stranded, negative-sense RNA (-ssRNA) divided into eight different segments. The viral RNA is packaged in viral ribonucleoprotein (vRNP) segments, each with its own coding RNA wrapped by the nucleoprotein (NP) and the RNA-dependent RNA polymerase (RdRp). Under the lipid envelope bilayer is the viral M1 (matrix 1) layer: an oligomeric lattice that constitutes the viral shell that protects vRNPs. The hypothesis—that vRNPs in virions retain characteristics of cellular condensates during entry and uncoating—is a valid one. There is evidence in IAV uncoating studies that the decondensation of incoming viral cores and viral RNA–protein complexes hijacks host processes linked to LLPS [118,119,120]. Studies on IAV entry and uncoating provide a clue that supports this hypothesis.

After binding to the cell surface sialic acids, IAVs are taken up into vesicles by receptor-mediated endocytosis. As the vesicle matures and its internal ionic environment shifts (increase in H+, K+), the viral core is exposed to the influx of such ions by virtue of the virally-encoded M2 (Matrix 2) ion channel [121,122]. The influx of H+ and K+ primes the viral core by breaking down M1–M1 oligomeric interactions and increasing the solubility of vRNP–vRNP complexes as shown by in vitro uncoating assays [123]. At low pH (<5.5), the hemagglutinin undergoes a conformational change that triggers membrane–membrane fusion of the viral and endosomal membranes [124,125]. Fusion is followed by viral core uncoating in the cytosol by host factors histone deacetylase 6 (HDAC6) and transportin-1 (TNPO1), among others. HDAC6 is a cytosolic deacetylase with ubiquitin-binding activity via a C-terminal Zinc-finger (ZnF)-ubiquitin binding domain (UBP) [126]. It is a cellular surveillance factor and master regulator of aggresome processing, which is a function that is important for the removal of misfolded protein aggregates from the cytosol by autophagosomal degradation [126,127,128,129,130]. HDAC6 ZnF is also essential for the assembly of inflammasomes [131]. Incoming IAV particles contain unanchored ubiquitin chains that mimic a misfolded protein aggregate which in turn activates HDAC6 and aggresome processing, promoting viral uncoating on the surface of late endosomes [118]. Interestingly, HDAC6 has been shown to promote LLPS by deacetylating the lysine residues of RNA helicase DDX3X IDRs [102]. 

TNPO1 is a nuclear import receptor of the importin beta family [132]. TNPO1 downregulates the cytosolic condensation of ALS protein FUS—an RNA-binding protein and transcriptional regulator—via the recognition of so-called unstructured PY-nuclear localization signal (NLS) sequences on FUS [44,133]. TNPO1 promotes the uncoating of IAV cores during cell entry, namely the decondensation of vRNP–vRNP complexes, via the recognition of non-canonical PY-NLS sequences on viral M1 [119]. TNPO1 preferentially binds acidified M1 dimers/oligomers instead of newly synthesized M1 [119]. A similar TNPO1-mediated uncoating mechanism was shown for the capsid (CA) protein of human immunodeficiency virus-1 (HIV-1), which is an enveloped +ssRNA virus of the Retroviridae [134] that causes LLPS during viral assembly and budding (see Section 2.4.2). During viral entry, TNPO1 uncoats incoming HIV-1 capsids via binding to a PY-NLS on the surface of hexameric CAs, but not monomeric CA [135]. The parallel observed between LLPS suppression and virus uncoating promoted by TNPO1 in two distinct enveloped RNA viruses (that are known to induce LLPS during assembly and budding) is striking. It supports the hypothesis that oligomers that constitute the viral shell hijack regulators of cellular condensates. The outcome is beneficial for the virus, as it facilitates uncoating, genome release, and the establishment of infection [120].

### 2.2. Formation of Replication Factories/Viral Inclusions

Studies have shown that *Mononegavirales* have broadly evolved mechanisms to assemble proteins into liquid compartments. To facilitate the description of how they are formed, we will introduce the common aspects of the viral structure and lifecycle of this order. The *Mononegavirales* order includes important pathogens of plants and animals. They are enveloped viruses carrying non-segmented genomes, which are made up of an -ssRNA [136]. The phylogenetic tree shows the eight current families of this order; they are *Bornaviridae*, *Mymonaviridae*, *Filoviridae*, *Nyamiviridae*, *Paramyxoviridae*, *Pneumoviridae*, *Rhabdoviridae*, and *Sunviridae* [136], in which there are many relevant pathogens to human health. Examples include MeV, parainfluenza virus (PIV), respiratory syncytial virus (RSV), RABV, mumps virus (MuV), human metapneumovirus (hMPV), and viruses causative of hemorrhagic fevers such as Ebola virus (EBOV), Marburg virus, or Nipah virus (NiV). They share a common structural element that is shared in the genomic composition of *Mononegavirales*, 3’-N-P-M-G-L-5’ (Figure 5), and they encode a variable amount of identified viral proteins (5-11) depending on the virus species. 

While N, M, and G code for nucleoprotein, matrix protein, and glycoprotein, respectively, the P (phosphoprotein) and L (Large protein) codify the RdRp [22,23,26,28]. During infection, these viruses are internalized by endocytosis and subsequently release their RNP into the cytosol, where most of these viruses form specialized viral replication sites called viroplasm or viral inclusions [137]. Viral inclusions, lacking delimiting membranes, are enriched for viral RNA, N, P, and L viral proteins, but how each of these proteins influences their formation differ in sufficiency and essentiality for different viruses, as explained below. Interestingly, despite having distinct amino acid sequences, these proteins share the common ability to segregate from the cytosolic milieu by LLPS, at least in vitro [22,23,26,28].

#### 2.2.1. RABV Replication Sites Are Liquid Organelles

The first example of a liquid compartment was provided for RABV. RABV is a neurotropic pathogen that causes fatal infections in both humans and animals, that although being vaccine-preventable, is reported to circulate in more than 150 countries. It causes severe progressive encephalitis, myelitis, and paralysis [138]. Its genome, of 12 Kb, is wrapped by the protein N in a complex with the RdRp L and its cofactor P. In the cytosol, vRNPs form specialized cytoplasmic viral inclusions, called Negri Bodies (NBs), of 2-10 μm in size, that serve as hubs for viral genome transcription and replication [22,139,140]. NBs are membraneless, and they are enriched for N, P, and L but also contain M, as well as the host proteins Hsp70 and focal adhesion kinase (FAK) [140,141,142]. Using a recombinant RABV expressing a fluorescent P protein, NBs were shown to behave similar to liquids, as they are spherical, exchange material dynamically, suffer fusion and fission events, and react to stimuli [22]. Overexpressing N and P proteins results in cytoplasmic spherical liquid inclusions that contain HSP70 and FAK, as those formed in infection [22]. However, they differ in the loss of the ability to eject material (presumably vRNPs) for posterior delivery to the plasma membrane [22]. The characteristics of N and P proteins enabling phase separation have not been fully mapped, but both proteins share properties common to those undergoing LLPS: N is an RNA binding protein and contains 30% IDRs. P contains IDDs spanning 75% of its sequence and a dimerization domain. Domains required for assembling liquid NBs in P protein were pinpointed to the N-RNA binding domain and IDD2, but the amino acids responsible for condensation are not known. Additionally, many questions remain to be solved, including how the material properties of NBs change over the course of infection or whether liquid NBs inhibit antiviral activation [139]. 

#### 2.2.2. VSV Replication Compartments Are Liquid Organelles

Liquid replication compartments were also described for VSV. VSV causes non-fatal infections in insects and mammals such as cattle, horses, and pigs with clinical manifestation identical to foot and mouth disease virus (FMDV) [143]. VSV breaches the host cell by using its G protein, a class III viral fusion protein, that mediates cell entry and membrane fusion with late endosome. Membrane fusion releases vRNPs consisting of the nucleocapsidated RNA and RdRp [144]. Replication compartments lacking delimiting membranes arise 4 h post infection to support VSV genome transcription and replication [145]. VSV genome replication requires newly translated N, P, and L proteins [146,147]. P protein, a component of VSV inclusions, has the characteristics of proteins able to achieve LLPS. It contains an oligomerization domain and highly disordered sequences but, despite this, P must interact with N and L to form VSV inclusions [145,148,149,150]. Active translation is essential for maintaining the phase separation of VSV inclusions, but *de novo* RNA synthesis is not required [23]. It is not known whether viral RNA changes the properties of liquid viral inclusions or if they mature over time, as for the case of NBs above. When N, P, or L proteins are expressed alone, only L forms VSV inclusions. However, inclusions with liquid properties are only formed in the presence of all three proteins, although N has been reported to have a smaller contribution to the fluidity of the structures [23]. 

### 2.3. Involvement of Liquid Compartments in the Formation of Replication Compartments and Genome Packaging

#### 2.3.1. The Case of MeV

A third example of a *Mononegavirales* member that forms liquid viral factories associated with viral replication without a delimiting membrane was reported for MeV [26,151]. MeV belongs to the genus *Morbillivirus* within the *Paramyxoviridae* family. It is a vaccine-preventable disease that worryingly is resurging on account of reduced vaccination. In 2018, 10 million cases worldwide were reported with 140,000 associated deaths [152]. Victims, especially children, die of complications (that are frequent in children under 5 years old or adults over 30), including pneumonia, diarrhea, and dehydration. However, MeV can leave permanent disabilities, including blindness, hearing loss, and brain damage. In addition, as it induces an immune amnesia, it can leave children vulnerable to other infections [152]. MeVs RNPs are composed of a 15.9kb -ssRNA associated with N, P, and L proteins, as in the preceding cases [153]. MeVs viral inclusions are enriched in RNAs, N, P, L, C [151,154], and host proteins such the WD repeat-containing protein 5 (WDR5) [151]. Similar to NBs and VSV, MeV liquid viral inclusions do not require the ongoing replication of RNA for formation [26], and N and P proteins, especially their C-terminal regions, are the critical components necessary for its biogenesis [26,28]. Using an in vitro system, Guseva et al. demonstrated that the critical components for LLPS are the amino acids 487-501 of the protein P, and that the mutation S491L in N suppressed phase separation [155]. The domain loop of P, which is an IDR, also contributes to phase separation by binding to N, as identified by NMR [26,28]. The ratio of condensed droplets showed dependence for a range of N concentration from 7.5 to 75 µM by a factor of about 3, which is indicative that the polyvalency of N:P interactions plays a role in droplet formation and maturation. Contributions of both disordered and multivalent domains stabilizing droplets invoke that these domains could behave as stickers and spacers described above, but the details need further investigation [38,70,73,156]. Interestingly, PTMs of the viral P protein were shown to be important for the biogenesis and physical properties of MeV inclusions, with the dephosphorylated form resulting in demixing of the structures. 

In terms of function in MeV infection, liquid inclusions were proposed to operate in transcription and replication, as their formation precedes and harbors these steps, and the S491L mutation has been known to lead to a reduction in viral transcripts [155]. It was speculated that this mutation could result in a decreased recruitment of the RdRp [157] promoted by N and P condensate formation. RNA preferentially localizes to condensates, triggering the assembly of nucleocapsid-like structures, in a reaction with an enhanced rate in relation to the observed in non-condensates [28]. Therefore, condensates may operate to coordinate several viral reactions, as will become clear in the next examples. 

#### 2.3.2. The Case of SARS-CoV-2

Coronaviruses (CoVs) belong to the family of *Coronaviridae*, the order *Nidovirales*, and the genus Coronavirus, and they have four genera: *Alphacoronavirus*, *Betacoronavirus*, *Gammacoronavirus*, and *Deltacoronavirus*. They encode a 30Kb +ssRNA genome, which is the largest known to date amongst RNA viruses. SARS-CoV-2 belonging to *Betacoronavirus* genera is a new zoonotic virus, which was first detected in humans in China in December 2019 [158]. It rapidly spread worldwide, becoming a pandemic. Therapies are urgently in demand to fight the excess death provoked by coronavirus disease 2019 (COVID-19). Viral replication occurs at the replication transcription complex (RTC) that is specifically formed during infection by reshaping ER membranes, giving rise to aggregated deconvoluted membranes mixed with double membrane vesicles [159]. SARS-CoV-2 genomic RNA must be coated by N before being selectively packaged into progeny virions [160]. N of human coronaviruses is highly expressed in infected cells. It is considered a multifunctional protein, promoting efficient sub-genomic viral RNA transcription, viral replication, virion assembly, and interacting with multiple host proteins [160,161,162]. 

Five studies have shown that the N protein undergoes LLPS in vitro [27,29,32,163,164] dependent on its C-terminal domain (CTD) [27]. The N protein of SARS-CoV-2 shares many features with proteins known to undergo LLPS: it is a multivalent protein and contains multiple RNA binding domains, several LCR, can oligomerize by N-N homotypic interactions, and scores as having high propensity to undergo LLPS by predictive tools [29,32,163]. The recombinant N-protein undergoes LLPS in vitro when above 50 µM, under physiological conditions, including pH, temperature, and ionic strength, and these parameters control the size and properties of condensates [27,32]. RNA facilitates the formation of bigger and more concentrated N-droplets [27,29], with high levels of RNA resulting in classical reentrant behavior [163,164]. It was also shown that the SARS-CoV-2 genome contains sequences and secondary structures able to promote LLPS (such as the 5’ end of genomic RNA) and able to inhibit condensate formation (such as within 1000 nt of Frameshifting-element of Orf1b), and those were proposed to, as a whole, provide an optimum material property for function [29]. Interestingly, Zn^2+^ favors N protein/RNA LLPS via oligomerizing the N-protein [27]. When overexpressed in cells, N forms liquid condensates that react fast to shock and stimuli [29]. 

How LLPS influences and impacts each of the functions of the N protein in SARS-CoV-2 infection is less well defined. It was hypothesized that by forming and/or being part of biomolecular condensates, the N protein of SARS-CoV-2 could not only build droplets to execute viral specific functions such as viral replication and assembly, but also shape the function of existing cellular biomolecular condensates, such as stress granules, and thus regulate host cell response to infection. In agreement, the N protein under conditions that do not favor LLPS was shown to enter into in vitro formed condensates of full-length human hnRNPs (TDP-43, FUS, and hnRNPA2) [32].

The N protein of SARS-CoV is thought to play a role in viral transcription and replication. Given that the N proteins of SARS-CoV and SARS-CoV-2 are 90% identical, one can predict that both proteins have a similar function [160,162]. By providing a cooperative mechanism to increase protein and RNA concentrations at specific localizations [18,74], the N protein could organize viral transcription and replication. In agreement, a recent study found that the replication machinery of SARS-CoV-2 was attracted to N protein/RNA condensates [163]. Using an in vitro system, the authors showed that recombinantly produced SARS-CoV-2 transcription machinery, composed of the non-structural protein (nsp) 12, together with the accessory sub-units nsp7 and nsp8 [160], colocalized with N protein/RNA formed condensates in vitro [163]. Supporting this idea, the N protein of the first SARS-CoV was shown to colocalize with replicase components [165] and RNA [166] in infected cells. 

In addition to its function in viral replication, studies proposed that the LLPS of N protein facilitated genome assembly and packaging of the 30 Kb genomic RNA of SARS-CoV-2 [27,29,32,164]. This is in line with reports on LLPS role in organizing genome packaging across the domains of life [167,168] and has been demonstrated for other viruses. For example, as mentioned above, the N and P proteins of MeV can undergo LLPS, leading to RNP condensation and the assembly of capsid-like structures [28]. This hypothesis also aligns with the need for electrostatic association of the positively charged N protein and the negatively charged RNA during genomic packaging [27]. Consequently, Cubuk et al. hypothesized that genome packaging into filamentous structures by the N protein [169] requires an initial nucleation process (driven by protein–protein and protein–RNA multivalent interactions), which precedes liquid-to-solid phase transitions as a way for the emergence of crystalline capsid structures [164]. 

A fascinating recent finding indicates that the phosphorylation of N could adjust the physical properties of condensates to facilitate the two different functions of N: transcription/replication and progeny genome assembly and packaging [29,163,170]. When N is phosphorylated, droplets have a liquid-like property. This characteristic could serve transcription/replication for many reasons. N phosphorylation is observed at the early stages of infection, a time at which transcription is occurring at the RTC that is rich in N [159,171]. In infected cells, N controls viral transcription when phosphorylated [172,173]. Such a fluid compartment is promoted by the phosphorylation of N and imposition of loose binding to RNA [163,170], where N can be linked to the viral replicase via the viral protein Nsp3 and this way enhance transcription. On the other hand, when N is unphosphorylated, N was shown to strongly associate with RNA by NMR, which could increase the strength of ligations in the droplets imposing liquid-to-gel-like transitions. In agreement, it was shown that at a fixed concentration of N, when N is unphosphorylated, RTCs are smaller, suggesting that unphosphorylated N promotes the maturation of droplets and leads to nucleopasid formation with a compacted RNA [163,170]. 

At the moment, the role of N-driven biomolecular condensates in infected SARS-CoV-2 cells is speculative and requires further investigation including evaluating its function in infected cells and consider how each component and respective modifications throughout the course of infection affect their physical properties and function. It is possible that N behaves differently in replication factories close to the ER or near the ER-Golgi intermediate compartment (ERGIC) where genome assembly is thought to take place. Despite the many open questions, the consistency in recent findings indicates that biomolecular condensates are functional in SARS-CoV-2 infection and pursuing this avenue may reveal novel aspects of virus-phase biology. 

### 2.4. Viral Assembly

#### 2.4.1. Assembly of Segmented Genomes: IAV

Genome packaging and virion assembly are vital aspects of all viral life cycles. This is a complex process for IAV due to the fact that its genome is segmented. The IAV genome contains eight different vRNPs [174,175,176]. Interestingly, most virions contain precisely eight segments and one of each type [175,176,177,178,179]. Therefore, a selective process has to ensure the fidelity of the IAV genome assembly [176,180,181,182,183,184,185,186]. 

Unusually for an RNA virus, influenza replication occurs in the host cell nucleus. vRNPs are subsequently exported to the cytoplasm. Evidence suggests that genome assembly takes place before reaching sites of virion assembly, which occurs at the plasma membrane. At the plasma membrane, complexes of eight interlinked vRNPs have been imaged (Figure 6) [175,183,187]. This means that at some point during infection, the eight segments assemble a higher-order-genomic-complex of IAV [177,179,188]. Many studies have suggested that the number of colocalizing different segments increased, while vRNPs were transported to the plasma membrane [177,179,189], aggregating in structures that increased in size [190,191,192,193]. These sites were hypothesized to contain partially assembled genomic complexes or sub-bundles. Recently, it was demonstrated that these sites have the properties of liquids as they suffer fusion and fission events, exchange material fast, coalescing into spherical-shaped structures, and react fast to several stimuli, including hypotonic shock and hexan-1,6-diol [24]. IAV inclusions are punctate cytoplasmic foci whose formation is driven by vRNPs as well as cellular host Rab11a GTPase, and develop close to ER exit sites [6,24,177,179,190,193,194,195,196]. Interestingly, it was observed that liquid viral inclusions are formed independently of interactions between different segment types, as cells transfected with a mini-replicon system expressing one single vRNP type assemble liquid structures (Figure 6) [24]. This observation raises the hypothesis that rather than being sites composed of partially or completely assembled genomes, liquid viral inclusions may be formed to facilitate the assembly of IAV genomes. Whether phase separation will serve to explain how segmented viral genomes self-assemble selectively is worth exploring [197,198,199]. Interestingly, LLPS was implicated in the formation of orderly assembled structures such as ribosomes, spliceosomes, microtubules, and actin [94,99,200,201,202]. It is also worth exploring whether liquid viral inclusions are sites for genome reassortment, with genome reassortment being the cause governing the emergence of pandemic IAV [199]. In this case, liquid viral inclusions could operate as determinants on viral host species jumps. This is an interesting hypothesis, given that the temperature of birds and humans varies and temperature is one of the factors that affects LLPS behavior and properties.

#### 2.4.2. Assembly and Budding of HIV

The human immunodeficiency virus, HIV, belongs to the Lentivirus genus and is a member of the retrovirus family. Upon infection, HIV can latently develop into AIDS in humans, which is a disease that subverts the immune system, leading to life-threatening opportunistic infections [203]. In 2019, 38 million individuals were estimated to live with HIV worldwide, and 0.7 million people died of AIDS. HIV genome is a dimeric positive sense single-stranded RNA (+ssRNA) of 9 Kb that contains nine ORFs, which produce 15 proteins [204]. It is enclosed within a capsid and a viral envelope derived from its host’s membrane. The viral lifecycle of HIV is complex, the +ssRNA is reverse transcribed to cDNA, the cDNA is integrated in the genome, and upon being transcribed, both mRNAs and genomic RNA are transported to the cytosol [205,206] to produce viral proteins Gag polyprotein precursor (also known as Pr55Gag), GagPol polyprotein precursor, the viral envelope glycoproteins (Env glycoproteins), and the regulatory and accessory viral proteins. Viral assembly is an orchestrated program regulated by Gag. Gag encapsidates the genomic RNA, creates a lattice at the plasma membrane that incorporates the viral Env glycoproteins, and initiates budding at the plasma membrane of infected cells [205,206,207,208]. The Gag precursor contains matrix (MA), capsid (CA), nucleocapsid (NC), p6 domains, and two spacer peptides, SP1 and SP2. Gag can operate as an uncleaved protein containing four domains that function as follows: the MA domain is responsible for delivering Gag to the plasma membrane and recruits the viral Env glycoproteins into forming virions; CA enables Gag multimerization and is essential for assembly; the p6 domain attracts the machinery responsible for fission and release of the virion—the endosomal sorting complex required for transport (ESCRT) apparatus, and NC recruits the viral RNA genome into virions [203,204,205]. 

For HIV genome assembly, dimeric viral RNA–Gag interactions result in the packaging of one RNA dimer per virion. NC contains two highly conserved ZnF domains responsible for many functions essential for the replication of the virus including binding nucleic acids (single-stranded DNA and RNA), as well as condensing and annealing them [209]. It was recently shown that in in vitro assays, NC as well as the uncleaved Gag can undergo LLPS, forming liquid structures that are spherical, divide, fuse, and the process is governed by the ZnF domain of NC [210]. It is accepted that proteins that undergo LLPS typically contain IDR or LCD and can bind DNA and RNA scaffolds through ZnFs or RNA-recognition motifs [85]. Interestingly, it is hypothesized that LLPS by NC in Gag can induce the repositioning of the viral genomic RNA and promote HIV genomic RNA packaging and trafficking to sites of virion assembly, and that this is a mechanism generally shared by retroviruses [211]. 

At the plasma membrane, it was also shown that the assembly and budding process of HIV is controlled by an active and sequential assortment into liquid-ordered lipids [134] as those observed for “lipid rafts” or cholesterol and sphingolipids-enriched domains at the plasma membrane [212]. At the site of viral assembly, Gag interacts with the plasma membrane through PIP2 [208] and oligomerizes [207] to initiate the local ordering of cholesterol and sphingolipids for the selective clustering of liquid-ordered GPI-anchored proteins (such as CD59, MLV envelope protein), while selective excluding proteins (such as GG, GT46) that segregate into the liquid disordered lipid phase [134]. It was hypothesized that lipid (dis)order facilitates the sequential redistribution and sorting of membrane proteins at assembly sites, including the exclusive late recruitment of important factors such as tetherin [213], by a mechanism that could rely on membrane curvature [134]. 

These findings broaden the applicability of liquid phase separation in modulating the life cycle of viruses (Figure 7). There are many viruses that bud at the plasma membrane exclusively on “lipid rafts” including IAV [214,215] that could use a similar mechanism to organize the composition of viral envelopes. 

## 3. How Liquid Organelles Participate in Viral Lifecycles: The Take-Home Message

Many viruses were shown to encode proteins able to undergo LLPS in in vitro reductionist models and when overexpressed in cells (Table 1). 

It is tempting to speculate that droplet formation serves a function and arises as a mechanism to facilitate the execution of specific steps in viral lifecycles and thus is shared amongst members of a viral family or an order (Figure 8). Consistently, other *Mononegavirales* such as *Filoviridae* form replication factories lacking delimiting membranes (Figure 3) and encode proteins able to undergo LLPS.

However, it may also be that droplet formation aroused independently and is a widespread phenomenon shared by unrelated viral families. In fact, the ability to undergo LLPS does not require conservation at the level of the amino acid sequence but rather summing a series of characteristics, including possessing IDR, LCR, and/or RNA binding domains. Interestingly, and in a broader sense, many unrelated viruses are capable of forming replication-like factories whose assembly is, at least partially, driven by their genome-coating proteins [23,28,31]. The next few years will certainly provide understanding on how LLPS operates in transcription/replication in viral-infected cells. Of note, LLPS-based mechanisms were shown to boost transcription in cells by multiple ways [60,217,218]. Finally, many other viruses were shown to form membraneless organelles. Examples include DNA viruses such as *Herpesviridae*, *Papillomaviridae*, *Adenoviridae*, *Parvoviridae*, and *Polyomaviridae* that replicate in the nucleus or *Poxviridae*, *Iridoviridae*, and *Asfaviridae* that replicate in the cytosol [13,31]. Whether any of these viruses assemble organelles with liquid-like behavior and whether LLPS is involved in their lifecycles remains to be elucidated.

## 4. Caveats of LLPS Studies in Virology

Studies exploring biomolecular condensates in viral infections have relied on two approaches: investigating if viral proteins undergo LLPS in minimalist in vitro assays, as seen for SARS-CoV-2 [27,32,163,170] or HIV [35], or using cells to provide descriptive accounts of the liquid behavior of biomolecular condensates [22,23,24,26,28,30,31,219], without understanding if the formation of these structures is governed by LLPS. In fact, liquid-like replication compartments built during Herpes Simplex Virus 1 behave in ways that are not consistent with LLPS principles [220]. Therefore, it is critical to examine and modulate the liquid compartments formed during viral infections to assess if LLPS or another interesting (and even uncharacterized) process governs their formation. Such requires building phase diagrams and determining exchange rates intra-condensates or between condensates and the milieu in physiologically relevant contexts [220,221,222]. This is important, because viruses can achieve similar end products by very diverse means. For example, it is well characterized that influenza mRNAs are poly-adenylated by a mechanism that greatly differs from the cellular polyA-tail formation [223]. Achieving such knowledge is critical to determine the significance of LLPS in virology. 

## 5. Perspectives and Final Remarks: Open Questions, Hypotheses, and Future Challenges

We are only beginning to decode the set of rules that govern the formation, maintenance, disassembly, regulation, and function of biomolecular condensates in biology. Viruses are well known for changing cellular structure and regulatory processes, and thus, it will be fascinating to provide mechanistic insight on the relation between viruses and the *de novo* formation of biomolecular condensates or the alteration of existing ones during infection. 

As viruses are obligatory parasites with no means of locomotion or energy production, it depends on the cell’s response to activate viral condensates to assemble new structures or to modify existing ones. How the cell coordinates these processes, either as a response to infection or with instructions supplied by virally encoded genes, is an interesting topic for the future. Condensates are a response to the concentration of a molecule crossing a critical separation threshold. Thus, molecular crowding in cells may dictate the efficiency of viral progeny production on a cell-to-cell basis, hypothesis for which the development of single-cell approaches could provide and answer. In addition, how the cell deals with the accumulation of viral proteins as infection progresses and how viral mutations affect the critical separation threshold has not been investigated. This notion is linked to the big question of how specific amino acids in proteins and RNA concentration tailor the material properties of biomolecular condensates and how the material properties (liquid, gel, or solid) suit specific viral-related functions. Perhaps this question may be answered by imposing material transitions to observe the impact in the efficiency of viral replication.

An additional unknown is how liquid organelles integrate into the structure of the cell. In particular, some papers have explored how viruses, e.g., IAV and SARS-CoV-2 assemble liquid biomolecular condensates associated with membrane organelles. Such association has been noted for other biomolecular condensates occurring in the cell and explored by other reviews [97,107]. 

Perhaps, though, the biggest challenge that this field faces is to provide evidence that LLPS plays a role in tissues and organs. Identifying ways to interfere with the formation and material properties of viral-induced biomolecular condensates in cells and test them in model organisms of infection may be useful to integrate LLPS into the physiology of organisms whilst exploring how that affects the outcome of viral infection. Viruses are composed of LLPS-prone components including RNA and globular proteins with unstructured/disordered domains that require assembly into spherical structures. It is no surprise that viruses have evolved to hijack the cellular LLPS machinery to streamline the production of their progeny. We envisage that by gaining a deeper understanding of how viruses hijack LLPS, we can, in turn, learn more about how our host cells regulate and manage biomolecular condensates. Such knowledge will connect disparate fields such as neurodegeneration and infection biology and present new opportunities that facilitate the development of therapeutic strategies. 

## Figures and Tables

**Figure 1 viruses-13-00366-f001:**
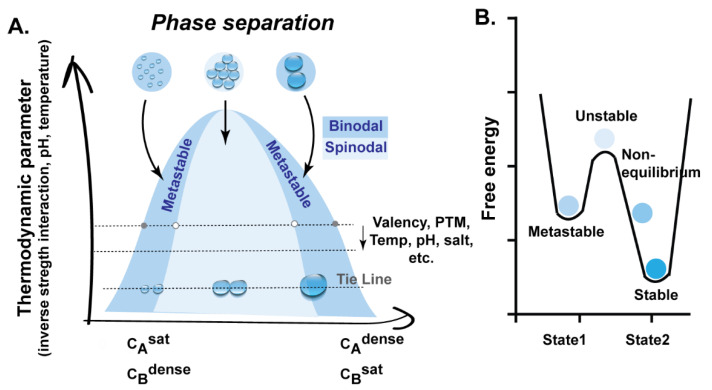
The principles of liquid–liquid phase separation (LLPS) in a binary system. Graphical representation of a phase diagram of a binary system (**A**) and how changes in energy drive the stability/instability of thermodynamic systems (**B**). The conditions leading to LLPS for a binary system composed of molecule A and solvent B (even if the solvent is in itself a mixture such as the cytosol) are depicted in the phase diagram in A. A phase diagram is a graphical representation of conditions (temperature, fraction of concentration of components, pressure, volume, pH, partition coefficient, inverse interaction strength) at which physical states of matter are thermodynamically stable. At very low concentration, A dissolves in B, mixing. As the concentration of molecule A increases, it reaches a point where the solubility of A in B saturates (at CAsat, concentration of saturation of A). Crossing this point, condensates start to form as follows: Molecules A and B can demix into two liquid phases when the binodal or coexistence line is reached. However, for this to happen, in the region between spinodal and binodal lines, a trigger or nucleation event must occur, and a metastable structure is obtained. As the concentration of molecule A increases, and the spinodal line is reached, the trigger is no longer necessary, as the mixture is unstable and spontaneously separates into two coexisting liquids or phases by a process called spinodal decomposition. A further increase in A, CAdense (dense phase concentration of A), passes the upper limit of the binodal line [48]. The mixture is energetically favorable, this time predominant in A. The tie line in the graph connects points of equal chemical potentials. In this line, the size of the droplets increases, but the concentration within the droplet is the same, because the partial molar free energy is equal. To change the molar fractions of CAsat and CAdense, the thermodynamic parameter must be changed. The lower the Csat (concentration of saturation), the stronger the driving forces of phase separation. At the critical point, the mixture is more favorable than the two existing phases [16,48,49]. Abbreviations: temp–temperature, PTM–post-translational modifications.

**Figure 2 viruses-13-00366-f002:**
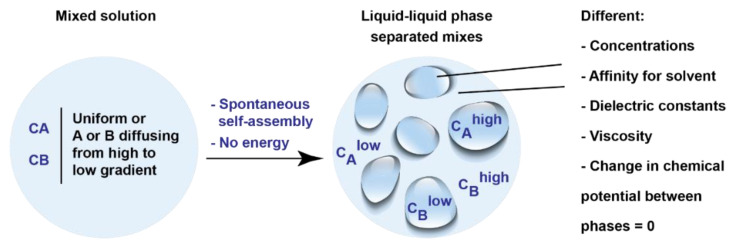
The consequences of LLPS in biological systems. CA, concentration of molecule A; CB, concentration of molecule B.

**Figure 3 viruses-13-00366-f003:**
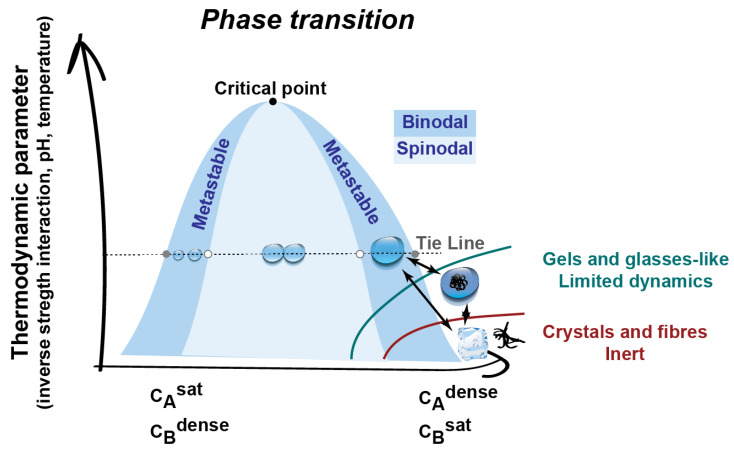
Phase diagrams with liquid, gels, and crystals. Biomolecular condensates can display a range of material properties from liquids with high dynamics to gels, hydrogels, and glasses that are less dynamic and to inert crystals or fibers as observed in neurodegenerative disorders. Phase transitions are possible by manipulating either the concentration, PTMs, valency, and strength of interactions (as happens in mutations) or by changing thermodynamic parameters such as ionic strength, temperature, pH, etc.

**Figure 4 viruses-13-00366-f004:**
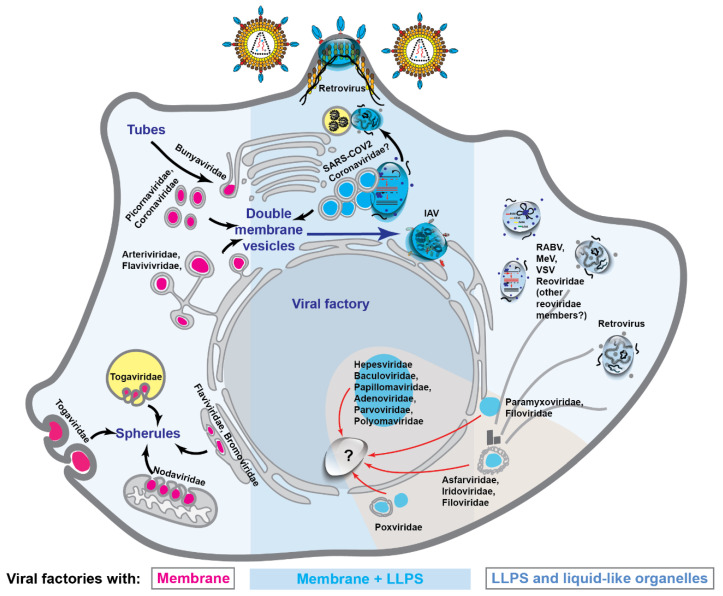
Viral infections reorganize cellular structure and organization. Seminal studies have shown several examples of viruses that re-shape membrane-bound organelles to facilitate their replication, which are depicted in the left side of the figure. Recent reports suggest that viruses may also induce the *de novo* formation of other types of cellular compartments, which lack delimiting membranes and display the properties of liquids, and that will be discussed in the next sections of this review. Interestingly, several viruses such as reovirus, influenza A virus (IAV), human immunodeficiency virus-1 (HIV), and severe acute respiratory syndrome coronavirus-2 (SARS-CoV-2) show an intimate association between the two types of cellular organelles, and the membrane interface has been shown to actively modulate protein LLPS in several ways. It has been shown to lower Csat for protein LLPS by restricting the movement of drivers and clients and even by operating as platforms for multivalent interactions. Additionally, it can build special environments to enable function; it can amplify signal transduction and even facilitate the fusion and fission of condensates, increasing their dynamics and avoiding that they reach equilibrium.

**Figure 5 viruses-13-00366-f005:**
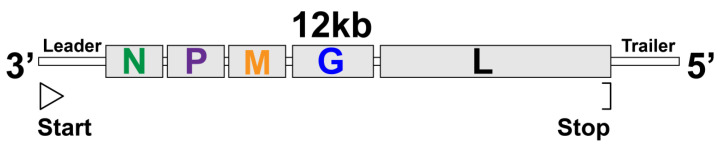
Graphical depiction of the genomic organization shared among the viral order of *Mononegavirales*. N, nucleoprotein; P, phosphoprotein; M, matrix protein; G, glycoprotein; L, Large protein.

**Figure 6 viruses-13-00366-f006:**
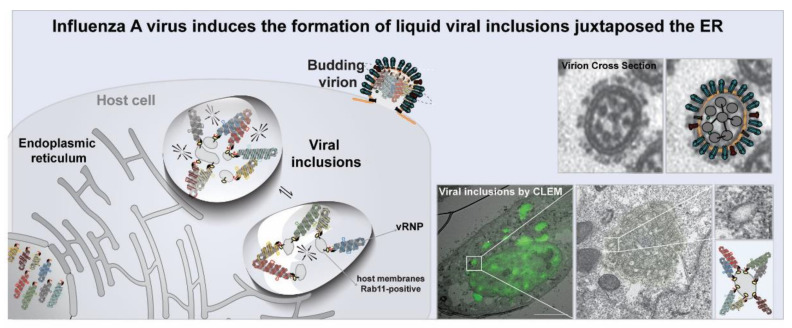
In IAV infection, viral ribonucleoproteins (vRNPs) concentrate in liquid compartments devoid of delimiting membranes—see correlative light and electron microscopy (CLEM) image—prior to be packaged into budding virions. CLEM also reveals membranes (inside) IAV liquid inclusions and electron dense material that resembles vRNPs in budding virions. Top right shows a cross-section of virions with eight vRNPs arranged as seven segments surrounding a central piece (reviewed in [190]).

**Figure 7 viruses-13-00366-f007:**
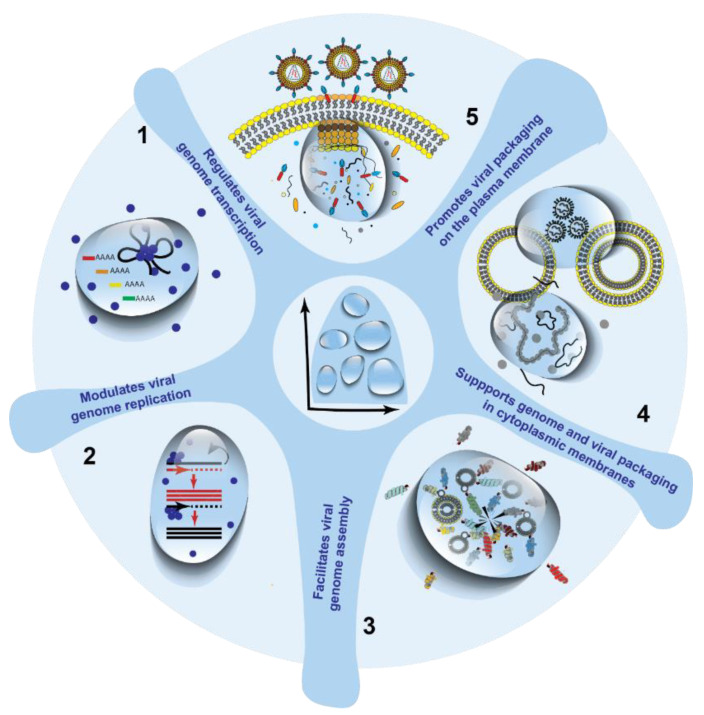
Liquid organelles can play different roles in viral infections. Despite being early days, liquid organelles have already been hypothesized to play many different functions in viral infections.

**Figure 8 viruses-13-00366-f008:**
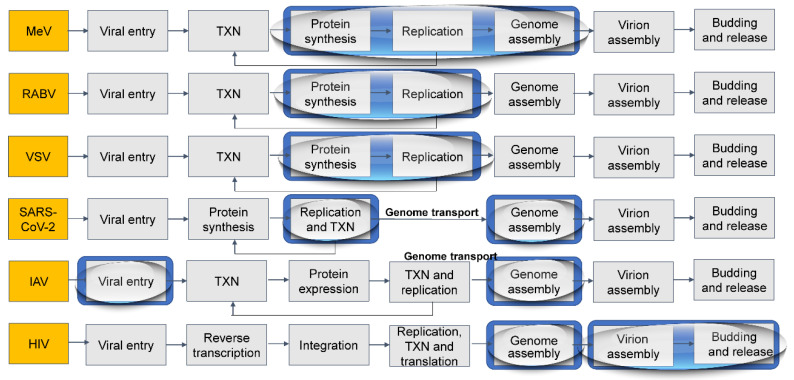
Graphical summary of the steps of the lifecycle of selected viruses, highlighted in the yellow box, with the inclusion, in blue, of the steps taking place in liquid organelles. Abbreviations mean transcription (TXN), measles virus (MeV), rabies virus (RABV), vesicular stomatitis virus (VSV), severe acute respiratory syndrome coronavirus-2 (SARS-CoV-2), influenza A virus (IAV), and human immunodeficiency virus (HIV). The graphical representation is inspired by adverse output pathways models.

**Table 1 viruses-13-00366-t001:** Summary of the viruses shown to use liquid organelles and the proteins necessary to induce their formation. Abbreviations: N–nucleoprotein; P–phosphoprotein; L–large protein, RNA dependent RNA polymerase; NP–nucleoprotein; vRNA–viral ribonucleoprotein; NC–nucleocapsid; (-)–not validated.

Virus	Liquid Compartment	Minimal Components	System	Validation	Ref
RABV	Negri body	N, P	Transfection	Infected cells	[22]
VSV	Viral Inclusion	P, (N, L)	Transfection	-	[23]
MeV	Viral factory	N, P	Transfection; In vitro reductionist assay	-	[26,28]
SARS-CoV-2	Replication transcription complex (RTC)	N	Transfection; In vitro reductionist assay	-	[27,29,32,163,170,216]
IAV	Viral Inclusion	NP, vRNA, Rab11	Transfection	Infected cells	[24]
HIV	Budding site	Gag/NC	Transfection In vitro reductionist assay	Infected cells	[35,134]

## Data Availability

The manuscript is a review of published literature and no data was acquired for it.

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
