# Peer review of "Liquid Biomolecular Condensates and Viral Lifecycles: Review and Perspectives"

_viruses, 2021, doi:10.3390/v13030366_

Round 1
Reviewer 1 Report
As a common mechanism for the assembly of membrane-less structures in a cell, liquid-liquid phase separation (LLPS) is emerging as a key process associated with the life cycle and pathogenesis of an increasing number of viruses. In this review, the authors explain the principles governing the formation and regulation of phase transition in general, followed by a discussion on recent progress and future perspectives on the potential roles of LLPS in the viral life cycle. The topic of this review is both timely and important. The manuscript is well organized and written, and the discussion is thoughtful and provocative. I only have a couple of very minor comments:
- On page 19, “During infection, these viruses are internalized by endocytosis….”: Some members of the Mononegavirales order, including RSV and MeV, can enter the cell via direct fusion on the plasma membrane.
- There appears to be citation program glitches at several places in the manuscript. For example, ref 24 should be cited instead of ref 120 for IAV on page 32. Additionally, ref 26 and ref 120 as well as ref 28 and ref 121 are redundant.
Author Response
We thank the reviewer for the comments. We have double checked and reformatted the entire bibliography. There were many mistakes, in fact and therefore this remark was highly valued as it has removed critical errors in the text.
Reviewer 2 Report
The authors explain and discuss the roles of liquid biomolecular condensates that forms membrane-less organelles in the infected cells with RNA virus. It is of general interest and has an impact in the field of virology as a whole, since biomolecular condensates and phase separation-driven cellular domains have been drawing a lot of attention lately. However, this review is a little bit verbose style. Therefore, re-organizing of some sections makes it more completed and readable.
1) Description regarding biomolecular condensates in DNA virus-infected cells is lacking. It would be better to state in the abstract that this review describes the phenomenon observed in RNA virus-infected cells.
2) The structure of some chapters seems inconsistent. For example, section1.4/1.5 and 1.3 (1.3.1 and 1.3.2). “Caveats and challenges of LLPS studies in virology” section looks like introduction. Therefore, re-organization is required.
Minor point:
3) there are some typos and incomplete citations.
P17 2nd paragraph; [145] (Ouyang et al., 2012, JBC)
P17 3rd paragraph; Transportin-1 (TNPO-1) (already mentioned in p17 2nd paragraph)
P32 line5; figure R
P32 line 8-9; [22-24, 26, Guseva. 2020, #750, 35, 261]
Author Response
We thank the reviewer for the comments. We have reorganized the following sections:
LIQUID COMPARTMENTS IN VIRAL INFECTIONS
Overview
- Virus entry and uncoating – the case of influenza A virus
- Formation of Replication factories/viral inclusions
- The case of SARS-CoV-2
HOW LIQUID ORGANELLES PARTICIPATE IN VIRAL LIFECYCLES: THE TAKE HOME MESSAGE.
CAVEATS OF LLPS STUDIES IN VIROLOGY:
PERSPECTIVES & FINAL REMARKS: OPEN QUESTIONS, HYPOTHESIS AND FUTURE CHALLENGES
Our alterations were quite exhaustive to make our text more completed and readable and to remove excess wording.
We have double checked the references and corrected all minor mistakes.
We believe that the changes have ameliorated the manuscript in ways that will please readers, the reviewer and the journal.